# Perceptions and Acceptability of a Low Phytate: Iron Molar Ratio Biofortified Bean and Sweet Potato Dish Among Pregnant Women in Rural Uganda

**DOI:** 10.3390/nu17101641

**Published:** 2025-05-11

**Authors:** Edward Buzigi, Kirthee Pillay, Muthulisi Siwela, Blessing Mkhwanazi, Mjabuliseni Ngidi, Babra Muhindo Mahinda

**Affiliations:** 1Department of Community Health & Behavioral Sciences, School of Public Health, College of Health Sciences, Makerere University, Kampala P.O. Box 7062, Uganda; 2Department of Dietetics and Human Nutrition, School of Agricultural, Earth and Environmental Sciences, University of KwaZulu-Natal, Private Bag X01, Pietermaritzburg 3201, South Africa; pillayk@ukzn.ac.za (K.P.); siwelam@ukzn.ac.za (M.S.); mkhwanazib@ukzn.ac.za (B.M.); ngidim@ukzn.ac.za (M.N.); 3Department of Public Health & Nutrition, Faculty of Health Sciences, Victoria University, Kampala P.O. Box 30866, Uganda; bmuhindo@vu.ac.ug

**Keywords:** pregnancy, iron deficiency, iron bean, acceptability, perceptions

## Abstract

**Background:** Iron deficiency anemia (IDA) disproportionately affects pregnant women who reside in low-income countries because they predominantly consume staple legumes and tubers such as conventional common beans (CCBs) and white-fleshed sweet potatoes (WFSPs). Such staples are either low in iron or rich in iron absorption inhibitors such as phytates. To fight IDA, a high-iron-biofortified common bean (IBCB) was introduced in Uganda. However, there is limited knowledge about its acceptability among pregnant women. This experimental crossover study determined the sensory acceptability of a low phytate:iron molar ratio dish, WFSP + IBCB (test food), against a high phytate:iron molar ratio dish, WFSP + CCB (control food), among pregnant women in rural Uganda. Their perceptions of consuming the test food during pregnancy were also explored. **Methods:** A total of 104 pregnant women participated in this study. The sensory attributes (taste, color, aroma, texture, and general acceptability) of the test and control foods were rated using a five-point facial hedonic scale ranging from “dislike very much”, “dislike”, “neutral”, “like”, to “like very much”. An attribute was acceptable if the participant scored either “like” or “like very much”. Focus group discussions (FGDs) were conducted to explore participant perceptions about the factors that may influence them to eat WFSP + IBCB during pregnancy. The chi-square test was used to detect the proportion difference for each sensory attribute within participants between test and control foods, while FGD data were analyzed by thematic analysis. **Results:** All the sensory attributes were acceptable to the participants and not significantly different between control and test foods (*p* > 0.05). Participants were willing to consume IBCB if it was affordable, sustainably available, and provided healthy pregnancy outcomes. **Conclusions:** The sensory attributes of the test food were equally accepted as the control food, suggesting that the consumption of WFSP + IBCB has the potential to replace WFSP + CCB among the study participants. The study participants showed positive perceptions of consuming IBCB if it was accessible, sustainable, affordable, and provided healthy pregnancy outcomes.

## 1. Introduction

Anemia occurs when the number of healthy red blood cells is insufficient to meet the body’s physiological demands for oxygen and nutrient delivery to the body’s tissues [1]. Hemoglobin is the primary oxygen-carrying molecule within red blood cells, hence the biomarker for measuring anemia [1]. Anemia in pregnancy, defined as hemoglobin of less than 11.0 g/dL, is a widely recognized nutrition-related preventable problem with serious consequences for pregnant women and their newborns because it is a leading risk factor for several adverse pregnancy outcomes such as maternal mortality, low birth weight, preterm birth, neonatal anemia, and neonatal deaths [2]. Globally, iron deficiency anemia (IDA) is the most common form of anemia that disproportionately affects pregnant women due to the increased physiological demands of iron required for increased red blood cell production and blood volume needed to meet the demands of fetal growth and development observed in pregnancy [3,4]. Moreover, the progress on anemia in women of reproductive age (WRA) aged 15–49 years, including pregnant women, is insufficient to meet the World Health Assembly’s global nutrition target to halve anemia prevalence by 2030 in low- and middle-income countries (LMICs), including Uganda [5,6].

Conventional common bean (CCB), *Phaseolus vulgaris*, is a staple legume usually consumed and available for consumption by pregnant women residing in LMICs such as Uganda [7,8]. However, CCB is either low in iron or rich in iron absorption inhibitors such as phytate and polyphenols that reduce iron bioavailability in consumers, including women [7,9]. Therefore, the low iron content coupled with high amounts of iron absorption inhibitors inherent in plant-source foods (PSFs) such as common beans has been linked to the 40−50% prevalence of IDA observed among pregnant women in Uganda [6,10]. To improve the iron content and reduce iron absorption inhibitors of CCB, iron biofortification of common bean programs was introduced in common bean-growing and consuming LMICs, including Uganda [7,11,12]. Iron biofortification of common beans is a nutrition-sensitive strategy that increases the concentration of iron in the edible portions of common beans through conventional breeding, fertilizer application, or bioengineering (recombinant DNA technology) to improve the iron status of target consumers [13]. Compared to other nutrition-specific strategies, such as industrial fortification and supplementation traditionally used to combat dietary iron deficiency (ID) and IDA, iron biofortification is relatively cost-effective because once biofortified seeds are released, the rural poor from LMICs can plant them season after season for sustained consumption [14,15].

The Recommended Dietary Allowance (RDA) is the average daily dietary intake level that is sufficient to meet the nutrient requirements of nearly all healthy individuals in a particular life stage and gender group. It is worth noting that the RDA for iron among pregnant women is higher at 27 mg/day compared to non-pregnant women at 18 mg/day [16]. To contribute towards achieving the RDA for iron among pregnant women, iron supplementation programs are recommended by the World Health Organization (2016) during the antenatal care (ANC) period [17,18]. However, adherence to iron supplementation during pregnancy is low, likely due to the related side effects such as constipation and undesirable sensory attributes that are reported by pregnant mothers, including those who reside in Uganda [19,20,21]. Therefore, a continued intake of heme iron-rich animal-source foods (ASFs) such as meat and organ meats would be necessary to improve the iron status of pregnant women [22]. However, ASFs are expensive for pregnant women who reside in LMICs [23]. Therefore, they resort to low-iron PSFs such as common beans, which contain high amounts of iron absorption inhibitors, including phytate and polyphenols [24,25].

To improve iron intake among vulnerable groups, such as pregnant women, the government of Uganda, through HarvestPlus, introduced iron biofortification of common beans [26]. Introducing and consuming IBCB is increasingly seen as a potential strategy for combating ID and IDA among vulnerable populations, such as pregnant women [27]. However, it is worth noting that understanding the acceptability of these biofortified foods is a prerequisite for the sustainable consumption of such newly released biofortified foods [28,29].

In Uganda, a staple dish prepared from a combination of WFSP and CCB (WFSP + CCB) is commonly consumed among WRA, including pregnant women [30]. However, a combination of WFSP prepared with CCB is low in iron compared to those prepared with IBCB [31]. If such a combination is habitually eaten, it has the potential to increase the risk of ID and IDA among WRA [25]. To contribute toward achieving the RDA for iron and combating IDA among pregnant women in rural Uganda, a homemade low phytate/iron molar ratio dish was prepared from WFSP and IBCB (WFSP + IBCB; test food) and compared to a conventional high phytate/iron molar ratio dish, WFSP + CCB (control food). It was plausible to prepare the test food because of its low phytate/iron molar ratio, which is associated with high iron absorption [12].

We hypothesized that the low phytate:iron molar ratio, WFSP + IBCB, and the conventional high phytate:iron molar ratio, WFSP + CCB, are equally acceptable to the study pregnant women. Therefore, this experimental study determined the sensory acceptability of a low phytate:iron molar ratio dish, WFSP + IBCB (test food), against a high phytate:iron molar ratio dish, WFSP + CCB (control food), among pregnant women in rural Uganda. Their perceptions of consuming test food during pregnancy were also explored.

## 2. Materials and Methods

### 2.1. Study Setting and Design

This experimental cross-over sensory acceptability study was conducted in Bwera General Hospital, in Kasese district, Western Uganda. This acceptability study was conducted in the Kasese district because it is located in the Tooro region, which is reported to have over 50% anemia prevalence among WRA [32]. Bwera Hospital was specifically chosen because it is the highest-volume health facility in the district, receiving over 500 pregnant women for ANC per month.

### 2.2. Study Participants and Sample Size Determination

A sample size of 50 participants or more is considered adequate for a valid cross-over sensory acceptability study [33,34]. The study participants were pregnant women attending ANC services in August 2023 at Bwera General Hospital in rural Western Uganda. A total of 104 pregnant women participated in the cross-over sensory acceptability study.

### 2.3. Sampling Procedure

The study participants were recruited by a systematic random sampling method. In a systematic random sampling, a number is assigned to every record, and then every subject is selected at a predetermined interval from a list [35]. On average, 40 pregnant women come for ANC services in Bwera General Hospital every other day. Therefore, every morning, a list of the first 40 pregnant women who came for ANC services was obtained from the ANC register. We systematically recruited 12 of 40 pregnant women per day in the ANC clinic between the 4th and 25th of August 2023. Therefore, we divided 40 by 12 to obtain an interval of 3. A number from 1 to 3 was chosen at random as the starting point of recruitment. In this case, number 3 was chosen. Therefore, every 3rd pregnant woman was recruited in this study. Pregnant women numbered 3, 6, 9, 12, 15, 18, 21, 24, 27, 30, 33, and 36 from the ANC register were recruited daily until 104 pregnant women were chosen.

### 2.4. Inclusion and Exclusion Criteria of Study Participants

A study participant was included in this study if she was pregnant. A pregnant woman was excluded from this study if she had any form of illness or her pregnancy was in the first trimester. This exclusion criterion was necessary because several illnesses may affect sensory attributes such as smell and taste in pregnant women [36]. Pregnant women in the first trimester were excluded from this study because severe nausea and vomiting (hyperemesis gravidarum) are highly prevalent in the first trimester [37,38,39]. Therefore, this could lead to an inaccurate sensory evaluation of the attributes by pregnant women in the first trimester of pregnancy.

### 2.5. Pilot Study

The pilot study was conducted to test the feasibility of methods and procedures for the preparation of study foods, sensory evaluation, and focus group discussions (FGDs) for later use in the main study. The village health team members identified seven expert peer pregnant women from the community to participate in the preparation and cooking of the study foods. The expert pregnant women cooked common beans and WFSP using boiling and steaming, as explained elsewhere [31].

Twenty pregnant women participated in the sensory evaluation pilot study. A short while later, all 20 pregnant women who had participated in the sensory evaluation participated in the pilot FGDs. Two FGDs were conducted during the pilot study, and each focus group had 10 pregnant women. The pilot study was conducted one week before the main study. This pilot study was conducted on a different day from the main study to prevent the pilot study participants from participating in the main study. The pilot study established that the majority (98%) of the pregnant women who participated in the pilot study had low literacy levels (did not complete primary school education). This was confirmed by records from the ANC clinic. Knowing the literacy level of the potential study participants guided the main study in modifying the sensory evaluation hedonic scale from a seven-point scale to a five-point scale. It is worth noting that longer hedonic scales, such as those with 7 or 9 rating scales, tend to confuse participants with lower literacy levels, while rating scales that are shorter than the five-point scale tend to cause end-point avoidance [33]. As the pilot study venue was too far away from the ANC clinic, a closer alternative venue was identified for the main study. No changes were made to the focus group discussion guide after the pilot study. The pilot study was also used to train research assistants.

### 2.6. Ingredients for the Preparation of the Study Composite Dishes

Composite foods are characterized by being multi-ingredient and include both ready-to-eat products and home-prepared dishes [40]. This study used home-based cooking methods commonly used to prepare common beans and sweet potatoes in Uganda. The control composite dish was WFSP served with CCB (WFSP + CCB). This combination was selected because it is a non-biofortified dish habitually consumed in Uganda, and it is characterized by low iron content [31]. The test composite dish was WFSP served with IBCB (WFSP + IBCB). The test composite dish was selected because it had a high IBCB [31]. Figure 1 shows the ingredients used to prepare the control and test composite dishes for the acceptability study.

### 2.7. Preparation of Test and Control Composite Dishes

The village health team members identified five expert peer pregnant women from the local community and invited them to the Bwera Hospital kitchen to participate in the preparation of the study composite dishes. Expert peer pregnant women were encouraged by research assistants (bachelor nutritionists) and village team members to prepare the composite dishes using locally acceptable home-based methods used in their households to prepare common beans and sweet potatoes to be consumed by pregnant women. We procured the IBCB (*NARO bean 4c*) from the National Crops Resources Research Institute, Namulonge, Uganda [41]. The commonly consumed WFSP (*Ebiribwa*) and CCB (*Nambale*) in the community were procured from the local market by expert peer pregnant women who prepared the composite dishes used in this study in the presence of the research assistants. The expert peer-pregnant women prepared common beans and sweet potatoes separately using local home methods of boiling and steaming, respectively, as explained in the previous Ugandan study [31].

### 2.8. Iron, Polyphenol, Phytate Composition, and Phytate: Iron Molar Ratio of Cooked Study Common Bean

The iron concentration of the common bean was analyzed by flame atomic absorption spectroscopy as described elsewhere [31,42]. The total polyphenol content in common beans was determined by spectrophotometry using the Folin–Ciocalteu method as explained elsewhere [43]. The anion-exchange method was used for the determination of the phytate content, as explained by Ma and colleagues [44]. Triplicate analysis for each cooked common bean variety was conducted to obtain an average of each iron, polyphenol, and phytate content. The phytate/iron molar ratio was significantly lower in IBCB compared to the CCB. Table 1 shows the iron, polyphenol, and phytate composition and phytate/iron molar ratio of the cooked IBCB and CCB used in the preparation of WFSP + IBCB and WFSP + CCB, respectively.

### 2.9. Measurement of Study Variables

#### 2.9.1. Measurement of Sensory Acceptability

Sensory acceptability was measured by the sensory evaluation method as explained elsewhere [34]. The sensory evaluation sessions were held in a room with separate, isolated cubicles set up for each panelist. Each panelist received a spoon and a small polystyrene cup containing 50 g of the control composite dish, WFSP + CCB, and 50 g of the test composite dish, WFSP + IBCB. To prevent pregnant women from making judgments based on labels, the dish samples were given random numbers as labels. The purpose of this was to ensure that panelists relied solely on their sensory experiences to assess the samples [34]. To achieve this, the samples were randomly labeled with a unique three-digit code obtained from a table of random numbers and were served in a random sequence [34].

The samples were warmed in a microwave oven for 10 seconds on medium heat before serving. The pregnant women were provided with a cup of water to rinse their palates between evaluating samples. Before each session, the sensory attributes of color, texture, aroma, taste, and overall acceptance were explained to the pregnant women. An equally spaced five-point facial hedonic scale with ratings (1 = “dislike very much”; 2 = “dislike”; 3 = “neither like nor dislike”; 4 = “like”; and 5 = “like very much”) was used. Each sensory attribute was described on the form with an accompanying facial hedonic scale. The ratings of the hedonic scale were verbally explained to the pregnant women in the local language during the sensory evaluation sessions. The participants were asked to rate the acceptance of each attribute by marking the appropriate response on the facial hedonic scale. A sensory attribute was considered acceptable if it was rated as either “like” or “like very much.” Figure 2 shows the flow diagram for the sensory acceptability study.

#### 2.9.2. Measurement of Pregnant Women’s Perceptions

The FGDs were conducted to explore the pregnant women’s perceptions to understand the factors that may motivate them to consume the test composite dish. The FGDs were conducted about 30 minutes after the sensory evaluation study was completed. Through established community relationships, four facilitators, experienced in conducting FGDs, were recruited for two days of training in the moderation of FGDs, following guidelines explained elsewhere [45]. A trained facilitator led the FGDs using a structured guide. The open-ended question was, “You have participated in the sensory evaluation of WFSP combined with IBCB. What would motivate you to consume the WFSP + IBCB compared to the WFSP + CCB?”. Furthermore, the facilitators used probes such as “Would you explain further?” and “Would you give an example?” when it was deemed necessary. The FGDs were facilitated in the local language (*Lhukonzo*) by trained facilitators. A digital voice recorder was used to record the FGDs. The FGDs were conducted until data saturation was reached. The focus group was composed of 8 to 10 participants. Data saturation was reached on the ninth focus group discussion, when we did not find any additional or new data from the FGDs [46].

### 2.10. Data Analysis

#### 2.10.1. Analysis of Quantitative Data Generated from Sensory Acceptability Measurements

Statistical data analysis was conducted using STATA, version 15.1. A sensory attribute was considered acceptable if rated as either “like” or “like very much” by the pregnant women. We created a binary outcome of yes/no for the sensory acceptability variable. A chi-square test was used to test for significant differences in the proportion of the binary outcome within participants between the test and control foods. The chi-square test was considered significant at a *p*-value of less than 0.05.

#### 2.10.2. Analysis of Qualitative Data Generated from Focus Group Discussions

Qualitative data generated from FGDs were analyzed using inductive thematic analysis by following the six steps, including familiarization of data, coding of data, developing themes, reviewing themes, defining and naming themes, and writing up, as shown in Appendix A [47].

### 2.11. Ethical Considerations

This study was performed following the ethical standards as laid down in the 2024 version of the Declaration of Helsinki. Ethical approval was granted by the AIDS Support Organisation Research Ethical Committee (reference number, TASO-2023-252). Informed and signed consent was obtained individually from the pregnant women attending the postnatal clinic of Bwera General Hospital, Kasese district, Western Uganda. Informed consent was taken from legally authorized representatives and/or guardians of all participants who were below 18 years old and those without formal education. The FGD guide consisted of a brief explanation of the samples that were evaluated during the sensory evaluation session, as well as the questions for initiating and facilitating the discussion. Guidelines for conducting FGDs with a structured set of open-ended questions were followed, as recommended elsewhere [45].

## 3. Results

### 3.1. Background Characteristics of Study Participants and Composite Dishes

A total of 104 eligible pregnant women participated in both the sensory evaluation study and FGDs. Their mean age and standard deviation were 24.5 years and ±2.6, respectively. Only 29% of the pregnant women had at least completed primary education, while 47% and 53% of the study participants’ pregnancies were in the second and third trimesters, respectively. Table 2 shows the background characteristics of the study participants.

### 3.2. Sensory Acceptability of the Study Composite Dishes

A sensory attribute was considered acceptable if the pregnant woman scored “like much” to “like very much” on the five-point hedonic facial scale. A binary outcome for sensory acceptability (yes or no) was created for each sensory attribute. Table 3 shows the association between each sensory attribute of the study composite foods and sensory acceptability.

Out of the 104 pregnant women, 103 (99%), 101 (97%), 98 (94%), 94 (90%), and 100 (96%) scored color, texture, aroma, taste, and general acceptability of WFSP + IBCB as acceptable (from “like much” to “like very much”), respectively. In contrast, out of the 104 pregnant women, 102 (98%), 101 (97%), 101 (97%), 96 (92%), and 99 (95%) scored color, texture, aroma, taste, and general acceptability of WFSP + CCB as acceptable (“like much” and “like very much”), respectively. The chi-square test revealed no significant difference in pregnant mothers’ sensory acceptability for all sensory attributes between WFSP + IBCB and WFSP + CCB (*p* > 0.05).

### 3.3. Perceptions on the Consumption of Study Composite Foods

Data saturation was achieved during the ninth focus group discussion. Six key themes emerged, reflecting issues that may explain pregnant women’s willingness to consume WFSP + IBCB. The themes included sensory acceptability, feasibility to prepare, availability, affordability, sustainable supply, nutritional value, and health benefits of consuming IBCB with either WFSP or any other staple foods.

#### 3.3.1. Sensory Acceptability Attributes

The vast majority of the pregnant women in the FGDs reported no barriers to choosing either WFSP + IBCB or WFSP + CCB for sensory reasons because both dishes were almost similar in all sensory attributes of color, aroma, texture, and taste.


*“I thought that both dishes were composed of WFSP and CCB. If you had not said that one dish was for WFSP + CCB and the other was for WFSP + IBCB, I wouldn’t have noticed that the two varieties of common bean in either WFSP + IBCB or WFSP + CCB were any different in color, texture, smell, and taste.” (A woman at 6 months gestation)*


The majority of pregnant women could hardly believe that IBCB was not similar to CCB in all the sensory properties.


*“…. it is hard to accept that IBCB is different from CCB because both have a similar smell, colour, taste, and texture.” (A woman at 4 months gestation)*



*“.. you wonder why sometimes health workers do not want to tell us the truth. How could someone say that IBCB is a different variety from CCB, yet both varieties are similar in smell, colour, taste, and texture.” (A woman at 8 months gestation)*



*“Someone may wonder why we are wasting time on this. We all tasted, touched, saw, and smelled on both IBCB and CCB. I could not find any difference in the two common bean varieties in terms of smell, colour, texture, and taste.”*


#### 3.3.2. Feasibility to Prepare Iron-Biofortified Common Bean for Household Consumption

The pregnant women noted they would use the IBCB if they cooked more quickly than the CCB and if they had adequate time and fuel to cook them. Some pregnant women indicated some of the preparation techniques for reducing the cooking time of the common bean.


*“… one major problem with common beans is that they take a long time to cook. You end up using a lot of fuel. I would like to know whether this IBCB variety takes a shorter period to get ready after cooking compared to CCB.” (A woman at 6 months gestation)*



*“…. I hate cooking common bean because it consumes a lot of fuel and time to become ready. I would only use IBCB if it cooked faster than the CCB.” (A woman at 8 months gestation)*



*“We realized that IBCB was similar to CCB in all the sensory properties. It is likely that they also have similar cooking times. Trust me, IBCB should have a long cooking time like CCB.” (A woman at 8 months gestation)*



*“Common bean will always be common bean, whether IBCB or CCB variety. All varieties of common beans consume a lot of fuel and hence take a long time to cook. I will never cook common beans unless I feel I have excess firewood to waste.” (A teenager at 8 months gestation)*



*“… buying a half kilogram of common bean is cheaper (3000 Uganda shillings) than buying a half kilogram of beef (6500 Uganda shillings). However, cooking common beans is more costly than cooking beef because you will use fuel worth 5000 Uganda shillings in the former and 1000 Uganda shillings in the latter. Moreover, you will cook common beans for more than 3 hours compared to beef, which can be cooked between 30 to 45 minutes. If you calculate the fuel and time needed to prepare common beans compared to meat, common beans are too expensive.”*



*“… you see when you are pregnant, you need to cook fast and rest. However, cooking common beans takes a longer time. This will prevent me from cooking common beans.” (A prime gravida teenager at 6 months gestation)*



*“… no one should ever deceive you about the cooking time of the common bean. All common bean varieties, whether IBCB or CCB, take a long time to cook, nearly 4 to 5 hours to cook. However, there are some techniques that you can use to make common beans softer so that they can cook faster. For example, you can soak them in water overnight for about 8 hours. Trust, me, soaked beans will cook faster for 2 hours compared to the non-soaked beans, which may cook for 4 to 5 hours.” (A gravida 4 at 7 months gestation)*



*“Besides soaking in water, there is a certain type of salt called “ekisula”. This type of salt softens the common bean. You just get a pinch of salt, “ekisula”, you put it the boiling common bean, within an hour the common beans will be ready for human consumption.” (A gravida 3 at 4 months gestation)*


#### 3.3.3. Availability of Iron-Biofortified Common Beans in the Market and Local Farmers

The vast number of pregnant women showed interest in knowing either the market where they could purchase the IBCB or the local farmers in the district who grow IBCB.


*“……… tell us where that IBCB variety can be available to buy it. Are the IBCB varieties available in the same market where we buy the CCB?” (A prime gravida at 6 months gestation)*



*“Are these IBCB grown here in our district? If so, please let us know the specific part of the district. I am asking all this because if it is grown within our district, it can be easy to buy directly from the local farmers. You see, buying common beans from local farmers can be cheaper compared to someone who buys from the retail market.” (A gravida 2 at 7 months gestation)*


#### 3.3.4. Affordability of Iron-Biofortified Common Bean

The study participants expressed concern about the affordability of IBCB. The vast majority noted that they would use IBCB for household consumption if it was cheaper to purchase from the local market compared to CCB or if the cost of producing IBCB from their gardens was reduced.


*“How much is the cost of IBCB compared to CCB? Is IBCB cheaper than CCB? I would buy for my household consumption if it is offered at an affordable price from the local market.” (A woman at 4 months gestation)*



*“… We are subsistence farmers, and most of the food for our household consumption is accessed from our garden. I need to know the cost of the IBCB seeds so that we can buy them for planting season after season. I will plant them if they are either cheaper than the CCB or they are of the same cost as CCB.” (A woman at 8 months gestation)*



*“…. growing IBCB in our gardens would be the best option. However, it might be expensive in the long run if it is either easily attacked by pests and diseases or not drought-resistant. To what extent are the IBCB drought and pest-resistant compared to our local CCB?” (A woman at 7 months of gestation)*



*“… I guess IBCB are genetically modified foods (GMFs). You know, GMFs, including IBCB varieties, are not like our local CCB varieties. I suspect these IBCB varieties are easily attacked by pests compared to our CCB. This means that one will incur more costs on buying pesticides if you grow the IBCB.” (A woman at 5 months gestation)*


#### 3.3.5. Sustainable Supply of Iron-Biofortified Common Bean

Several pregnant women raised concerns about the sustainable supply of the IBCB in the food supply chain.


*“….. I would consume the IBCB if I could find it any time I needed it. The challenge is to go to the market and you don’t find the IBCB any time you may need to cook it for household consumption.” (A woman at 5 months gestation)*



*“.. I remember the Ministry of Agriculture officials sensitized us about the benefits of consuming IBCB. This happened about three years before I migrated to this district. However, I could not find the IBCB on the local market whenever I wanted them. It would be important that as you introduce these IBCB to us also put effort into ensuring that they are always available on the market season after season so that people can either consume them or grow them in their farm household.” (A woman at 8 months gestation)*



*“… health workers and agricultural extension workers always tend to introduce biofortified foods to us. However, these biofortified foods only appear once on the market when their funded project or programme is running. The sad reality is that when the programme ends, you will never find these biofortified foods again in the local community. I doubt whether this IBCB will be available in our community season after season.” (A woman at 5 months gestation)*


#### 3.3.6. Nutritional Value and Health Benefits for Pregnancy Outcomes

Several pregnant women suggested that they would consume IBCB if they knew that its consumption would provide better pregnancy and birth outcomes compared to CCB.


*“Are there any health benefits of consuming IBCB during pregnancy? I will only consider eating IBCB combined with WFSP or any other staple cereal or tuber if I know that IBCB has more health benefits for my fetus and myself than CCB.” (A prime gravida at 6 months gestation)*



*“I don’t see any difference in the sensory attributes between WFSP + IBCB and WFSP + CCB. Therefore, I would only choose to eat WFSP prepared with IBCB when I am sure that IBCB will improve birth outcomes of my newborn, such as a good childbirth weight and my general good health.” (A woman at 6 months gestation)*



*“… I think IBCB has nutrients that have health benefits, such as the formation of blood for the pregnant woman and the fetus. Before I migrated to this district, I was sensitized that IBCB has a certain nutrient that helps in the formation of blood. However, I have forgotten the name of the nutrient. Since a pregnant woman needs a lot of blood, I would consume IBCB during pregnancy to increase the amount of my blood.” (A woman at 5 months gestation)*


## 4. Discussion

To the best of our knowledge, this is the first study to assess perceptions and sensory acceptability of IBCB homemade composite dishes prepared from iron-rich IBCB among pregnant women in Uganda and other countries where IBCB has been released [29,48]. In the present study, the sensory acceptability attribute ratings were equally high and not significantly different among participants for both the test and control foods. Furthermore, pregnant women had positive perceptions of consuming WFSP + IBCB if IBCB was sustainably supplied, affordable, and accessible, and its consumption provided healthy pregnancy outcomes.

Some of the sensory acceptability findings observed in this study have been observed elsewhere among non-pregnant WRA [29,31,49,50]. A recent study conducted among lactating mothers in Uganda demonstrated that IBCB prepared with a provitamin A-rich orange-fleshed sweet potato (OFSP) was equally acceptable as the CCB prepared with WFSP in sensory attributes of aroma, taste, and general acceptability, but not in color and texture [31]. In the study that involved lactating mothers, the color of the OFSP + IBCB dish was significantly preferred compared to WFSP + CCB, and this observation was attributed to the attractive yellow/orange color of OFSP [31]. In contrast, the texture of the WFSP + CCB dish was significantly more accepted by lactating mothers compared to OFSP + CCB, and this observation was attributed to the soft texture of OFSP, which the study participants did not like [31]. To this end, the differences in sensory acceptability attributes observed between the lactating mothers’ study [31] and our present study could be explained by the OFSP and WFSP used in the test composite dishes of the lactating mothers’ study and our study, respectively. In Rwanda, the sensory attributes for the two IBCB varieties were rated higher than those of the CCB among consumers of common beans, including WRA [49]. In Guatemala, common bean consumers, including women, equally preferred the IBCB and CCB varieties, although some minor differences were found in some of the sensory attributes [50]. It is worth noting that the sensory acceptability results observed in the previous studies should be interpreted with caution because these earlier studies were conducted in non-pregnant women [29,31,49,50,51]. In contrast, our study was conducted among pregnant women, demonstrating a broader applicability in a group most vulnerable to sensory acceptability changes [37,52] and IDA [3,4].

Findings from the sensory acceptability study suggest that the consumption of the iron-rich and low phytate/iron molar ratio, WFSP + IBCB, has the potential to replace the low-iron and high phytate/iron molar ratio, WFSP + CCB, in the study pregnant women. Moreover, the consumption of foods prepared with IBCB with a low phytate/iron molar ratio has been demonstrated to improve iron status biomarkers and reduce IDA among WRA [53]. Therefore, consuming IBCB would complement other nutrition interventions such as iron supplementation and enriching the dietary diversity recommended during pregnancy to improve iron status and achieve positive pregnancy outcomes [18].

Our findings that emerged from qualitative data align with previous studies conducted among non-pregnant women in countries where biofortified food crops, including IBCB, have been released [31,54,55]. In Uganda, lactating mothers emphasized that they would consume IBCB combined with provitamin biofortified sweet potato if they were affordable in terms of fuel savings and market price, accessible, sustainable, and provided health benefits to them [31]. In South Africa, caregivers expressed a willingness to give their infants porridge made with provitamin A-biofortified maize if it was more affordable, readily available, and beneficial to health [54]. Another South African study found that caregivers agreed to feed their children the provitamin A biofortified meals because they were culturally acceptable [55]. Findings from our FGDs and other qualitative studies [31,54,55] inform the need to sensitize potential consumers, including pregnant women, about the nutritional value and health benefits of consuming IBCB [50]; ensure a sustainable supply of biofortified foods in such a subsistence farming community through food production diversity [56]; and release easy-to-cook IBCB varieties by the biofortification experts [57].

### Study Strengths and Limitations

Several strengths are inherent in this study. Compared to other studies that measured the acceptability of biofortified foods based on only quantitative data generated from sensory acceptability hedonic scales [29,51], our study used qualitative (FGDs) and quantitative (sensory acceptability) data collection methods. Qualitative data helped validate sensory acceptability findings through triangulation. For example, the sensory acceptability quantitative data showed that both test and control foods were equally acceptable to the pregnant women. Moreover, in the FGDs, the pregnant women highlighted the reasons for the sensory acceptability. Furthermore, measuring perception provided an opportunity to understand the other non-sensory acceptability factors that would influence pregnant women to accept consuming IBCB. Compared to a sensory acceptability study of a biofortified food that used two parallel groups of study participants [58], ours used a crossover design with the following advantages. The sensory acceptability was compared among the study participants since each received both test and control foods. This removed the inter-participant variability from the comparison between the test and control food groups and the effect of confounders that can be observed in studies with parallel groups of participants [59].

Some limitations also exist in our study. Compared to the 7- or 9-hedonic scales, the 5-hedonic scale offers limited freedom for participants to express a wider range of their hedonic experiences [60]. However, using a shorter 5-point hedonic scale was necessary because most of our study participants had lower literacy levels (over 71% had not completed primary education). It is worth noting that longer hedonic scales, such as those with 7 or 9 rating scales, tend to confuse participants with lower literacy levels, while rating scales that are shorter than the 5-point scale tend to cause endpoint avoidance [33]. Moreover, the qualitative part of our study provided participants with the freedom to express their opinions on what would influence them to accept consuming the IBCB.

## 5. Conclusions and Recommendations

In conclusion, we can accept the hypothesis that the sensory attributes of the low phytate:iron molar ratio IBCB served with WFSP were equally accepted as the high phytate:iron molar ratio CCB served with WFSP, suggesting that the consumption of WFSP + IBCB has the potential to replace WFSP + CCB among the study participants. Pregnant women showed positive perceptions of consuming IBCB if it was accessible, sustainable, affordable, and provided healthy pregnancy outcomes. Therefore, we recommend that nutrition-sensitive agriculture programs should consider sustainably growing IBCB in this rural district, where the majority of the study participants are subsistence farmers. Future studies should investigate the acceptability of IBCB served with other staple foods (cereals, tubers, and plantains), followed by intervention studies to evaluate the effect of IBCB intake on iron status biomarkers and pregnancy outcomes in this study population.

## Figures and Tables

**Figure 1 nutrients-17-01641-f001:**
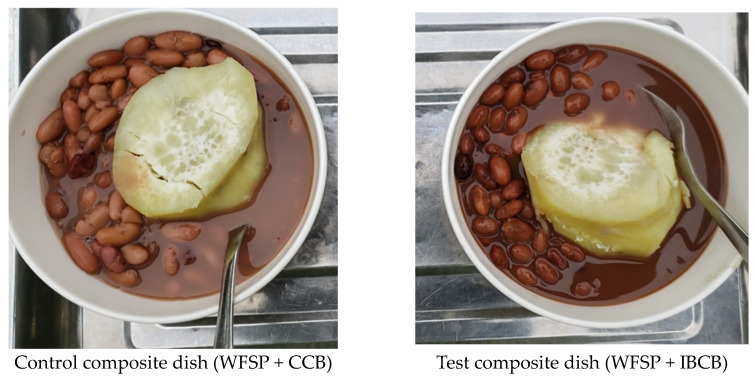
Test and control composite dishes prepared from sweet potato and common bean. WFSP + CCB: A combination of white-fleshed sweet potato and conventional common bean. WFSP + IBCB: A combination of white-fleshed sweet potato and iron-biofortified common bean.

**Figure 2 nutrients-17-01641-f002:**
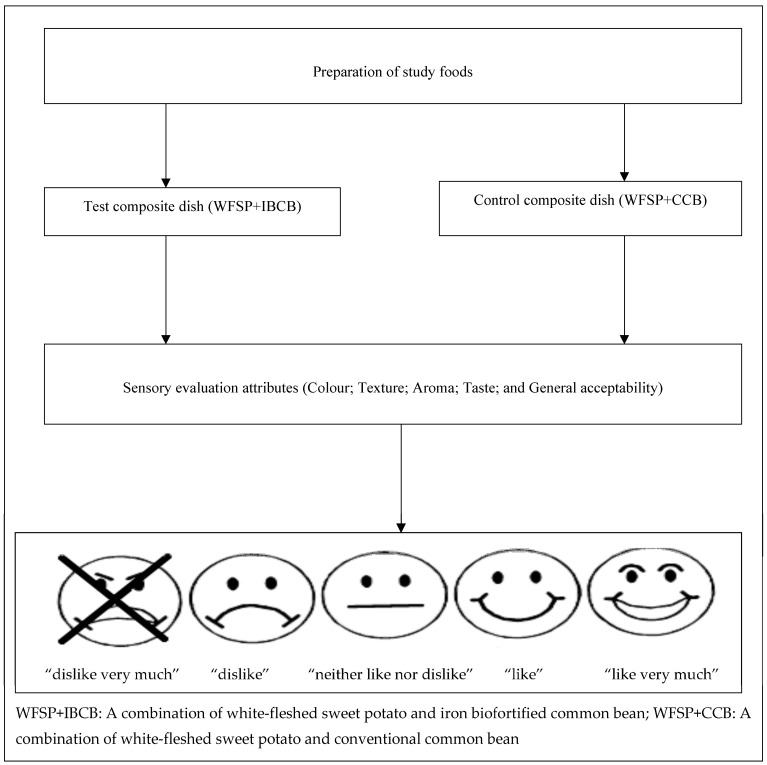
Flow diagram for the pregnant women’s sensory acceptability study.

**Table 1 nutrients-17-01641-t001:** Iron, polyphenols, phytate, and phytate/iron molar ratio of IBCB and CCB.

Variable (Units)	IBCB/100 g ^$^	CCB/100 g
Iron (mg, mean ± SD)	9.5 ± 0.3 ^a^	5.1 ± 0.2 ^b^
Total polyphenols (mg GAE, mean ± SD)	572 ± 10 ^a^	453 ± 12 ^b^
Phytate (mg, mean ± SD)	1321 ± 10 ^a^	982 ± 20 ^b^
Phytate: iron molar ratio	12:1 ^a^	16:1 ^b^

CCB: conventional common bean; IBCB: iron-biofortified common bean; mg: milligram; g: grams; GAE: garlic acid equivalent; SD: standard deviation. Values in the same row with different superscript letters are significantly different (*p* < 0.05). ^$^ The iron biofortification target for common bean, ≥9.4 g iron/100 g, was achieved [14].

**Table 2 nutrients-17-01641-t002:** Background characteristics of the 104 pregnant women who participated in the sensory acceptability study and FGDs.

Variable	Frequency	Percentage (%)
**Adolescent**		
Yes (10–19 years old)	24	23.1
No (≥20 years old)	80	76.9
**Completed primary education**		
Yes	30	28.8
No	74	71.2
**Trimester**		
Second	49	47.1
Third	55	52.8
**Household head**		
Male	89	85.6
Female	15	15.4
**Subsistence farmer**		
Yes	91	87.5
No	13	12.5

**Table 3 nutrients-17-01641-t003:** Association between sensory attributes and sensory acceptability of study composite foods among 104 pregnant women.

Sensory Attribute	Acceptable		Chi-Square	*p*-Value
	Yes, *n*(%)	No, *n*(%)		
Color			0.338	0.56
WFSP + IBCB WFSP + CCB	103(99.0) 102(98.1)	1(1.0) 2(1.9)		
Texture			0.000	1.00
WFSP + IBCB	101(97.1)	03(2.9)		
WFSP + CCB	101(97.1)	03(2.9)		
Aroma			1.045	0.307
WFSP + IBCB	98(94.2)	06(5.8)		
WFSP + CCB	101(97.1)	03(2.9)		
Taste			0.243	0.62
WFSP + IBCB	94(90.4)	10(9.6)		
WFSP + CCB	96(92.3)	08(7.7)		
Overall acceptability			0.116	0.73
WFSP + IBCB	100(96.2)	04(3.8)		
WFSP + CCB	99(95.2)	05(4.8)		

WFSP + IBCB: A combination of white-fleshed sweet potato and iron-biofortified common bean; WFSP + CCB: A combination of white-fleshed sweet potato and conventional common bean.

## Data Availability

The datasets used and/or analyzed in the present study are available to the corresponding author on reasonable request. The data are not publicly available due to privacy reasons.

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
