# Peer review of "Perceptions and Acceptability of a Low Phytate: Iron Molar Ratio Biofortified Bean and Sweet Potato Dish Among Pregnant Women in Rural Uganda"

_nutrients, 2025, doi:10.3390/nu17101641_

Round 1

Reviewer 1 Report

Comments and Suggestions for Authors

This paper is promising in perceptions and acceptability of a composite dish prepared  
from high iron, and low phytate. However, it needs to be revised before it can be accepted. (1) It is suprising that this paper has 69 references for an article. (2) The quality of figure 2 needs to be improved. (3) What is the advantages and disadvantages of this method? (4) The results in this method should be compared with other methods.

Author Response

(1) It is suprising that this paper has 69 references for an article.

Agreed: This article had many references because the authors did an extensive literature review and added many explanations. However, the extensive explanations have been reduced, hence reducing the references from 69 to 61

(2) The quality of figure 2 needs to be improved.

Agreed: A new high-quality figure is now provided.  See figure 1, line 190.

(3) What is the advantages and disadvantages of this method?

Agreed: The advantages and disadvantages of the methods are now explained under section “4.1 Study strengths and limitations”

(4) The results in this method should be compared with other methods.

Agreed: These are now discussed under section 4.1, Study strengths and limitations.  Other studies with different methodologies are now compared in this section 4.1.  See lines 567 to 582

Reviewer 2 Report

Comments and Suggestions for Authors

Thank you very much for allowing me to review the manuscript entitled “nutrients-3573264_Perceptions and acceptability of a composite dish prepared from high iron, and low phytate: iron molar ratio iron biofortified common bean and white-fleshed sweet potato among pregnant women in rural western Uganda“ which has been submitted to the “Micronutrients and Human Health” of this journal.

This study evaluates the perceptions and sensory acceptability of a high-iron, low-phytate: iron molar ratio dish comprising white-fleshed sweet potato (WFSP) and iron-biofortified common bean (IBCB) (test food) compared to a low-iron, high-phytate: iron molar ratio dish, WFSP with conventional common bean (CCB) (control food), among pregnant women in rural Uganda.

Comments:

  1. The title is excessively long. I suggest: “Perceptions and Acceptability of an Iron-Biofortified Bean and Sweet Potato Dish Among Pregnant Women in Rural Uganda.” This revised title retains the essential information while being more concise and comprehensible. The authors may consider an alternative shorter title if deemed appropriate.
  2. The keywords are overly lengthy phrases. I recommend using MeSH (Medical Subject Headings) classification to facilitate better indexing of the manuscript.
  3. The introduction effectively outlines the issue of anaemia during pregnancy in low- to middle-income countries, detailing the needs of pregnant women and the potential benefits of the proposed intervention. The references cited are appropriate. However, the introduction should conclude with a clear statement of the study’s objective rather than discussing its potential utility. The authors may also consider explicitly stating the hypothesis before presenting the objective in a precise manner.
  4. The Materials and Methods section describes the study as a cross-sectional design conducted in 2016. Given that we are in 2025, this may be an error. Furthermore, the ethical approval is dated 2023, which raises concerns regarding consistency. Please clarify this discrepancy.
  5. Based on the stated objective in the abstract, the study compares two groups subjected to different dietary interventions, which suggests an experimental rather than a purely cross-sectional design. This aspect should be clarified.
  6. Was a sample size calculation performed? If so, this should be explicitly stated in the Materials and Methods
  7. The article by Hough et al. (2006) [Hough, G.; Wakeling, I.; Mucci, A.; Chambers IV, E.; Gallardo, I.M.; Alves, L.R. Number of Consumers Necessary for Sensory Acceptability Tests. Food Qual. Prefer. 2006, 17, 522–526, doi:10.1016/j.foodqual.2005.07.002] suggests an N value of 112 consumers for similar parameters, which should apply to each group being compared. Please clarify this issue.
  8. Line 126 states that participants were recruited through systematic sampling; however, the methodology is not clearly explained. Please specify the systematic sampling approach used and the source list. Additionally, the inclusion and exclusion criteria should be detailed. It is crucial to demonstrate that there are no significant differences between the two groups receiving different dietary interventions.
  9. How many women declined participation? What type of personnel was responsible for teaching the participants how to prepare the food?
  10. Does the study assess culinary preparation or sensory perception?
  11. Table 1 presents the two food preparation methods under comparison; however, no statistical significance tests are provided. These should be included to support the claims of nutritional or sensory benefits of the proposed dish.
  12. In the Results section, line 368 reports the mean age but does not provide the standard deviation. A table should be included detailing the baseline characteristics of the two groups to confirm the absence of significant differences.
  13. Tables 3 and 4 contain identical content. They should be merged into a single table. Furthermore, the comments presented in the results section should not be interpreted as results; their inclusion should be justified.
  14. The discussion describes this as an exploratory study. If this is the case, the title, study design, methodology, and results should reflect this to ensure consistent interpretation.
  15. The study’s limitations should be explicitly discussed.
  16. The conclusion should directly answer the research objective.

I appreciate the opportunity to review this manuscript and look forward to the authors’ revisions.

Author Response

  1. The title is excessively long. I suggest: “Perceptions and Acceptability of an Iron-Biofortified Bean and Sweet Potato Dish Among Pregnant Women in Rural Uganda.” This revised title retains the essential information while being more concise and comprehensible. The authors may consider an alternative shorter title if deemed appropriate

    Agreed, the previous title, “Perceptions and acceptability of a composite dish prepared from high iron, and low phytate: iron molar ratio iron biofortified common bean and white-fleshed sweet potato among pregnant women in rural western Uganda” was longer.

    The title is now shortened and rewritten as “Perceptions and acceptability of a low Phytate: iron molar ratio Biofortified Bean and Sweet Potato Dish Among Pregnant Women in Rural Uganda”  

  2. The keywords are overly lengthy phrases. I recommend using MeSH (Medical Subject Headings) classification to facilitate better indexing of the manuscript.

    Agreed: The Keywords are now shortened as: “pregnancy; iron deficiency; iron-bean; acceptability; perceptions”

  3. The introduction effectively outlines the issue of anaemia during pregnancy in low- to middle-income countries, detailing the needs of pregnant women and the potential benefits of the proposed intervention. The references cited are appropriate. However, the introduction should conclude with a clear statement of the study’s objective rather than discussing its potential utility. The authors may also consider explicitly stating the hypothesis before presenting the objective in a precise manner.

    Agreed: We now include the hypothesis as “We hypothesized that the low phytate: iron molar ratio, WFSP+IBCB, and the conventional high phytate: iron molar ratio, WFSP +CCB, are equally acceptable to the study pregnant women”.  Lines 109-111

    The objective  is now restated in a more precise manner as “Therefore, this experimental study determined the sensory acceptability of a low phytate: iron molar ratio dish, WFSP+IBCB (test food), against a high phytate: iron molar ratio dish, WFSP+CCB (control food), among pregnant wom-en in rural Uganda. Their perceptions of consuming the test food during pregnancy were also explored. See lines 111-115.

    4. The Materials and Methods section describes the study as a cross-sectional design conducted in 2016. Given that we are in 2025, this may be an error. Furthermore, the ethical approval is dated 2023, which raises concerns regarding consistency. Please clarify this discrepancy.

    No, our study was not a cross-sectional design.  This may have been misread. In section “2.1, study setting and design”, it was stated that “This experimental cross-over sensory acceptability study was conducted in Bwera General Hospital, in Kasese district, Western Uganda.”

    Furthermore, our study was not conducted in 2016.  The “2016” is the year when the most recent Uganda Demographic Health Survey was conducted. This citation was used to justify that the prevalence of anemia among women of reproductive age, including pregnant women, in the study setting was unacceptably high at over 50%. So, the 2016 citation was used to justify that it was necessary to conduct this acceptability study in such a study area with a high prevalence of anemia.

    Furthermore, the ethical approval is dated 2023, which raises concerns regarding consistency. Please clarify this discrepancy.

    The ethical approval for the study was granted in 2023.

    To clarify this discrepancy, we have revised section 2.1: study setting and design as

    “This experimental cross-over sensory acceptability study was conducted in Bwera General Hospital, in Kasese district, western Uganda. It was necessary to conduct this acceptability study in Kasese district because it is located in the Tooro region, which is reported to have over 50% anemia prevalence among WRA [31]. Bwera Hospital was specifically chosen because it is the highest-volume health facility in the district, receiving over 500 pregnant women for ANC per month.” 

    5. Based on the stated objective in the abstract, the study compares two groups subjected to different dietary interventions, which suggests an experimental rather than a purely cross-sectional design. This aspect should be clarified. 

    Yes, this was an experimental study and not a cross-sectional study. 

    In section 2.1, Study setting and design, we wrote that “This experimental cross-over sensory acceptability study was conducted in Bwera General Hospital, in Kasese district, western Uganda”.  The word “cross-over” may have been misread as “cross-sectional.” 

    In the cross-over study, participants are exposed to both the intervention and control. In the context of our study, the pregnant women were exposed to both the test food and the control food during the sensory acceptability evaluation.

6. Was a sample size calculation performed? If so, this should be explicitly stated in the Materials and Methods

No, we did not calculate the sample size; our sample size was guided by Lawless and Heymann (2010-page 93) and  Stone and Sidel (2004-page 185) who argue that a sample size of 50 or more participants is adequate for a sensory evaluation study where participants discriminate study samples. 

This is now stated under section 2.2 study participants and sample size determination as

A sample size of 50 participants or more is considered adequate for a valid cross-over sensory acceptability study” [33,34]. The study participants were pregnant women attending ANC services in August 2023 at Bwera General Hospital in rural western Uganda. A total of 104 pregnant women participated in the cross-over sensory acceptability study.”

  The full references of these books are provided here

 Lawless, Harry T, and Hildegarde Heymann. 2010. Springer Science and Business Media Sensory Evaluation of Food: Priciples and Practises. Food Science Text Series. Second Ed. New York, USA: Springer Science+Business Media. doi:10.1007/978-1-4419-6488-5.

Stone, Herbert, and Joel L Sidel. 2004. Sensory Evaluation Practices. Third Ed. ed. Steve L Taylor. California, USA: Elsevier Academic Press.

 7. The article by Hough et al. (2006) [Hough, G.; Wakeling, I.; Mucci, A.; Chambers IV, E.; Gallardo, I.M.; Alves, L.R. Number of Consumers Necessary for Sensory Acceptability TestsFood Qual. Prefer. 2006, 17, 522–526, doi:10.1016/j.foodqual.2005.07.002] suggests an N value of 112 consumers for similar parameters, which should apply to each group being compared. Please clarify this issue.

This author was wrongly cited. We intended to cite  “Lawless and Heymann (2010) and Stone and Sidel (2004)”,  who argue that a sample size of 50 or more is adequate for a crossover sensory evaluation study.

To this end, the citation under section “2.2 Study participants and sample size determination” is now revised. It now reads: “A sample size of 50 participants or more is considered adequate for a valid cross-over sensory acceptability study” [33,34]. The study participants were pregnant women attending ANC services in August 2023 at Bwera General Hospital in rural western Uganda. A total of 104 pregnant women participated in the cross-over sensory acceptability study.  

8.Line 126 states that participants were recruited through systematic sampling; however, the methodology is not clearly explained. Please specify the systematic sampling approach used and the source list. Additionally, the inclusion and exclusion criteria should be detailed. It is crucial to demonstrate that there are no significant differences between the two groups receiving different dietary interventions.

Agreed: The revised manuscript now states that systematic random sampling was used and that the source of the list was the ANC register. Please see section “2.3 sampling procedure”

The inclusion and exclusion criteria were explained in detail in section “2.4. Inclusion and exclusion criteria of study participants.”

It is crucial to demonstrate that there are no significant differences between the two groups receiving different dietary interventions.

Disagree: Please note that this was a crossover experimental study. It is worth noting that in a crossover study similar to the current study, the study participants are switched throughout to all the treatment groups (both test and control food) after a washout period (rinsing the palate between tasting the two foods). Therefore, it is not true that the two groups received different dietary interventions. Because the same set of participants is exposed to both the test and control foods, the advantage of cross-over studies is that participants act as their own controls.

9. How many women declined participation? What type of personnel was responsible for teaching the participants how to prepare the food?

There were no women who declined to participate.  Therefore, it was not reported on.

What type of personnel was responsible for teaching the participants how to prepare the food?

Expert peer pregnant women selected from the community prepared the study foods. They were taught through the pilot study under the guidance of research assistants and village health team members. 

This was stated under section 2.7. Preparation of test and control composite dishes by writing “The village health team members (community health workers) identified five expert peer pregnant women from the local community and invited them to Bwera Hospital kitchen to participate in the preparation of the study composite dishes. Expert peer pregnant women were encouraged to prepare the composite dishes using locally acceptable home-based methods used in their household to prepare common beans and sweet potatoes to be consumed by pregnant women.”

10. Does the study assess culinary preparation or sensory perception?

The study did not specifically target to assess culinary preparation or sensory perception. The study assessed general perceptions of motivators to consume the test food.  This was explained in section 2.9.2 as “The FGDs were conducted to explore the pregnant women's perceptions to understand the factors that may motivate them to consume the test composite dish.” See line 295-296

11. Table 1 presents the two food preparation methods under comparison; however, no statistical significance tests are provided. These should be included to support the claims of nutritional or sensory benefits of the proposed dish.

Agreed: The table has been revised and now values in the same row with different superscript letters are significantly different (p < 0.05). It also shows that the iron biofortification target for the IBCB was achieved.

12. In the Results section, line 368 reports the mean age but does not provide the standard deviation. A table should be included detailing the baseline characteristics of the two groups to confirm the absence of significant differences.

Agreed: The age and standard deviation are now presented as “Their mean age and standard deviation were 24.5 years and ±2.6, respectively.” See line 343,

A table should be included detailing the baseline characteristics of the two groups to confirm the absence of significant differences.

Agreed: A table showing background characteristics of the study participants is now provided as Table 3.  However, it does not show participant characteristics between the two study foods to confirm the absence of significant differences because this was a cross-over study. Please note that in cross-over studies, the study participants are switched throughout to all the treatment groups (both test and control foods). Being the same set of the participants, the advantage of cross-over studies is that patients act as their own controls. Therefore, there is no need to confirm the absence of significant differences between participants receiving either test food or control food.  This is plausible because the participants were exposed to both foods. Therefore, our interest was to determine the difference in rating sensory attributes within participants, not between participants. 

13. Tables 3 and 4 contain identical content. They should be merged into a single table. Furthermore, the comments presented in the results section should not be interpreted as results; their inclusion should be justified.

  1. Tables 3 and 4 contain identical content. They should be merged into a single table.

Agreed: Table 3 has been deleted and content merged. 

Furthermore, the comments presented in the results section should not be interpreted as results; their inclusion should be justified.

Agreed, comments such as those on the nutrient composition of the study foods have been deleted in the results section. They are now shown in Table 1.  

14. The discussion describes this as an exploratory study. If this is the case, the title, study design, methodology, and results should reflect this to ensure consistent interpretation.

Agreed: This was a mixed-methods study where we collected both qualitative data (perceptions) and quantitative data (sensory acceptability). Therefore, we used the word explored because we explored participants' perceptions of accepting/willingness to consume IBCB. Moreover, the word “perceptions is reflected in the title, methodology, and results. Furthermore, we use the word “determined” sensory acceptability of the study foods to reflect the quantitative data to inform whether the sensory acceptability was significantly different within participants.

15. The study’s limitations should be explicitly discussed

Agreed:  The study’s limitations are now discussed under section “4.1 Study strengths and limitations.” 

16. The conclusion should directly answer the research objective.

Agreed: The objective is given as: “Therefore, this experimental study determined the sensory acceptability of a low phytate: iron molar ratio dish, WFSP+IBCB (test food), against a high phytate: iron molar ratio dish, WFSP+CCB (control food), among pregnant women in rural Uganda. Their perceptions of consuming the test food during pregnancy were also explored.”

 To answer this research objective, section “5 conclusion and recommendation” states: “In conclusion, we can accept the hypothesis that the sensory attributes of the low phytate: iron molar ratio IBCB served with WFSP were equally accepted as the high phytate: iron molar ratio CCB served with WFSP, suggesting that the consumption of WFSP+IBCB has the potential to replace WFSP+CCB among the study participants. Pregnant women showed positive perceptions of consuming IBCB if it was accessible, sustainable, affordable, and provided healthy pregnancy outcomes.” 

Reviewer 3 Report

Comments and Suggestions for Authors

This study evaluates the perception and sensory acceptance of two types of foods, each with different amounts of iron and phytates, in 104 pregnant women from the second trimester onward.

This is a very real research study born from the need to meet the iron needs of a population susceptible to Uganda. Some minor aspects could improve the quality of the research.
1. On line 25, the authors indicate 104 pregnant women; however, on line 122, they indicate 103. This aspect should be clarified.
2. The large number of abbreviations makes understanding specific sentences extremely difficult. It would be advisable to use abbreviations only for highly repetitive terms.
3. Overall, the manuscript is long and overly explanatory. Please be specific in all sections, focusing on and reducing apparent aspects.
4. LIMCs? Line 59
5. Focus on the objective of the study.
6. It is not necessary to indicate the five-point facial expressions in the abstract…
7. Sampling procedure. The sample sizes for each experiment are not sufficiently clear. There is confusion regarding dates (lines 130, 2023, and 115, 2026). This section is confusing and not easy to understand. I recommend focusing on the writing, and eliminating superfluous data. The final sample sizes for each experiment are important.
8. Line 148. It is not necessary to explain what a pilot study is. Focus on this section. Make a brief outline of the meal preparation. It is not necessary to explain everything in excessive detail. There is a lot of confusion between the sample sizes of the pilot study and the final ones. It would be advisable to create an initial table with the sociodemographic characteristics of the study population.
9. The images have low resolution. Fig. 1
10. Section 2.7. Summarize and focus the writing in an impersonal style. NARO bean 4c? Line 212, because different tenses are used.
11. Table 1. The total number of polyphenols is indicated, but only one is indicated, which is GAE. Please explain this.
12. Line 242. Missing reference.
13. Section 2.9. Summarize and focus the writing.
14. Line 265. “Like” and “like very much”
15. FGDs?
16. Repetition of lhukonzo, line 311 and line 322. Reduce and focus on this section. The number of participants in this section is confusing; it is necessary to outline the sample sizes used.
17. Line 322 should be in the ethics section.
18. FGDs? Line 340. It is not necessary to explain apparent terms.
19. Line 357. The 1964 Helsinki Declaration is very old; there are more recent versions.
20. Section 3.1 explains the contents of Table 1; they should not be explained here.
21. Line 383. A high proportion? Indicate the percentage.
22. Table 4. P=0.000. This value should be reviewed since the data compared are the same.
23. All verbatim data indicated in the rest of the article should be included in supplementary tables to simplify the work.
24. the results are repeated in the discussion, and references are made to the tables. This should not be the case.

Author Response

  1. On line 25, the authors indicate 104 pregnant women; however, on line 122, they indicate 103. This aspect should be clarified.

    Agreed: Writing 103 was a typing error. The correct number of study participants was 104 as indicated in the abstract and table of results. 103 is now corrected to 104.

  2. The large number of abbreviations makes understanding specific sentences extremely difficult. It would be advisable to use abbreviations only for highly repetitive terms.

    Agreed: The number of abbreviations has been reduced, and only repetitive ones are maintained. For example, NaCRRI, the abbreviation for National Crops Resources Research Institute, WTP (willingness to pay) and HH (household) have been deleted.

  3. Overall, the manuscript is long and overly explanatory. Please be specific in all sections, focusing on and reducing apparent aspects

    Agreed:  Several aspects have been reduced in the methodology, for example, table 2 showing the steps of thematic analysis is now put in the supplementary tables. Several repetitions have been deleted, citations reduced from 69 to 61, and pages reduced from 21 to 20.  

    4. LIMCs? Line 59

    LMICs, an acronym for low and middle-income countries. It was given in full the first time it was used. See line 59

    5. Focus on the objective of the study.

    Agreed: we now focus on the objective of the study which is stated as Therefore, “this experimental study determined the sensory acceptability of a low phytate: iron molar ratio dish, WFSP+IBCB (test food), against a high phytate: iron molar ratio dish, WFSP+CCB (control food), among pregnant women in rural Uganda. Their perceptions of consuming the test food during pregnancy were also explored.” See lines 111-115

    6.  It is not necessary to indicate the five-point facial expressions in the abstract…

    Agreed: We have now deleted the numbers corresponding to the ratings on the five-point scale i.e. (1 = dislike very much, 2 = dislike, 3 = neutral, 4 = like, 5 = like very much).

    7. Sampling procedure. The sample sizes for each experiment are not sufficiently clear. There is confusion regarding dates (lines 130, 2023, and 115, 2026). This section is confusing and not easy to understand. I recommend focusing on the writing, and eliminating superfluous data. The final sample sizes for each experiment are important.

    Agreed: Section “2.2. Study participants and sample size determination,” has been revised as follows: “A sample size of 50 participants or more is considered adequate for a valid cross-over sensory acceptability study” [33,34]. The study participants were pregnant women attending ANC services in August 2023 at Bwera General Hospital in rural western Uganda. A total of 104 pregnant women participated in the cross-over sensory acceptability study.  

    8. Line 148. It is not necessary to explain what a pilot study is. Focus on this section. Make a brief outline of the meal preparation. It is not necessary to explain everything in excessive detail. There is a lot of confusion between the sample sizes of the pilot study and the final ones. It would be advisable to create an initial table with the sociodemographic characteristics of the study population.

    Agreed: The explanation of what a pilot study is has been deleted.

    Make a brief outline of the meal preparation. It is not necessary to explain everything in excessive detail. 

    Agreed. The meal preparation has been described briefly.  Please see section 2.7 Preparation of test and control composite dishes, lines 203 to 205.

    There is a lot of confusion between the sample sizes of the pilot study and the final ones.

    The sample size for the pilot study was 20, as explained in “Section 2.5. Pilot study.”. In contrast, the sample size for the final study was 104, as shown in “Section 2.2. Study participants and sample size determination.”

     It would be advisable to create an initial table with the sociodemographic characteristics of the study population.

    Agreed: Table 3 has been added. It shows the background characteristics of study participants.

    9. The images have low resolution. Fig. 1

    Agreed, new images for figure  with a high resolution are now provided.

    10. Section 2.7. Summarize and focus the writing in an impersonal style. NARO bean 4c? Line 212, because different tenses are used.

    Agreed, the impersonal style of writing is now avoided.  The sentence is now revised as “We procured the IBCB (NARO bean 4c), the National Crops Resources Research Institute Namulonge, Uganda

    11. Table 1. The total number of polyphenols is indicated, but only one is indicated, which is GAE. Please explain this.

    GAE is not a type of polyphenol. GAE is an acronym for garlic acid equivalent, as shown in Table 1. Please note, GAE is part of mg GAE, the units of measuring polyphenols.

    12. Line 242. Missing reference.    Agreed: The reference [34] is now included as “Sensory acceptability was measured by the sensory evaluation method as explained elsewhere [34].”                                                                                              13 . Section 2.9. Summarize and focus the writing.

    Agreed: several redundant statements and repetitions have now been deleted, including “It is worth noting that the WRA in the study district have lower literacy levels, with 67% of children dropping out of school before completing ordinary-level education [44]. This coupled with low literacy levels established during the pilot study, justifies the use of the five-point rating scale in the study participants [32].” ;  “A sample size for a FGD between 7 and 12 participants is appropriate for a qualitative nutrition-related study [46]. Therefore, each focus group included 8 to 10 participants.”; “Qualitative data based on perceptions were collected using FGDs and a FGD guide.” 

    14. Line 265. “Like” and “like very much”  
    Agreed; this is now corrected as A sensory attribute was considered acceptable if it was rated as either “like” or “like very much.”   See line 246
    15. FGDs?

    “FGDs” is focus group discussions (FGDs). It was given in full the first time it was used in “Section 2.5 pilot study.”  See line 155

    16. Repetition of lhukonzo, line 311 and line 322. Reduce and focus on this section. The number of participants in this section is confusing; it is necessary to outline the sample sizes used.

    Agreed: the repeated “ lhukonzo,”one in line 311 has been deleted.

    The number of participants in this section is confusing; it is necessary to outline the sample sizes used.

    Agreed: we now outline the sample size as “The focus group was composed of 8 to 10 participants. Data saturation was reached on the ninth focus group discussion, when we did not find any additional or new data from the FGDs [45].”

    17. Line 322 should be in the ethics section.

    Agreed: it has been shifted to the ethics section

    18. FGDs? Line 340. It is not necessary to explain apparent terms.

    Agreed: The apparent terms have been deleted.

    19. Line 357. The 1964 Helsinki Declaration is very old; there are more recent versions.                                             . Agreed:  We now refer to the most recent 2024 version of the Declaration of Helsinki. See lines 328-329                        20. Section 3.1 explains the contents of Table 1; they should not be explained here.                                                  Agreed: this content has been deleted                                                                                                                                 21 Line 383. A high proportion? Indicate the percentage. 

    This was deleted as advised by the other reviewers since it was a repetition of results in Table 4

    22. Table 4. P=0.000. This value should be reviewed since the data compared are the same.

    Disagree: The 0.000 is not a P value. Please note that this is the chi-square value. The P value is 1.00, indicating that the texture was not significantly different for both foods within participants.

    23. All verbatim data indicated in the rest of the article should be included in supplementary tables to simplify the work. 

    Agreed:  We have now moved Table 2 to a supplementary table. Table 2. Description of the six steps for inductive thematic analysis used in the study

    24. The results are repeated in the discussion, and references are made to the tables. This should not be the case.

    Agreed:  The repetitions of results in the discussion are now deleted.

    Furthermore, the references to the tables are now deleted.

Reviewer 4 Report

Comments and Suggestions for Authors

  1. Limited Novel Contribution: While the study addresses a relevant topic, it does not offer substantial new insights. The research largely reiterates existing knowledge about iron biofortification without introducing novel methods or perspectives. More in-depth exploration of the potential long-term health benefits of the biofortified dish is needed.

  2. Insufficient Analysis and Context: The article does not provide sufficient analysis of the broader implications of its findings. It lacks a detailed discussion of the nutritional aspects of iron-biofortified foods, their impact on the overall diet, and how this biofortification could be scaled or integrated into broader health interventions in rural populations.

Author Response

  1. Limited Novel Contribution: While the study addresses a relevant topic, it does not offer substantial new insights. The research largely reiterates existing knowledge about iron biofortification without introducing novel methods or perspectives. More in-depth exploration of the potential long-term health benefits of the biofortified dish is needed.

    Agreed. In the discussion, we explore the potential long-term health benefits of IBCB   based on sensory acceptability and FGDs results

    See line 534 -541 where we write “Findings from sensory acceptability suggest that the consumption of the iron-rich, low phytate: iron molar ratio, WFSP+ IBCB, has the potential to replace the low-iron, high phytate: iron molar ratio, WFSP+CCB, in the study pregnant women. Moreover, the consumption of foods prepared with IBCB with a low phytate: iron molar ratio has been demonstrated to improve iron status biomarkers and reduce IDA among WRA [53]. Therefore, consuming IBCB would complement other nutrition interventions such as iron supplementation and enriching the dietary diversity recommended during pregnancy to improve iron status and achieve positive pregnancy outcomes [18].

    Also see lines 551 -556 were we write “Findings from our FGDs and other qualitative studies [31,54,55], inform the need to sensitize potential consumers, including pregnant women, about the nutritional value and health benefits of consuming IBCB [50,56]; Ensure a sustainable supply of biofortified foods in such a subsistence farming community through food production diversity [57], and the release of easy-to-cook IBCB varieties by the biofortification experts [58].

  2. Insufficient Analysis and Context: The article does not provide sufficient analysis of the broader implications of its findings. It lacks a detailed discussion of the nutritional aspects of iron-biofortified foods, their impact on the overall diet, and how this biofortification could be scaled or integrated into broader health interventions in rural populations.

    Agreed: These have been addressed in section 5, " Conclusion, and recommendations,” as follows:

    “In conclusion, we can accept the hypothesis that the sensory attributes of the low phytate: iron molar ratio IBCB served with WFSP were equally accepted as the high phytate: iron molar ratio CCB served with WFSP, suggesting that the consumption of WFSP+IBCB has the potential to replace WFSP+CCB among the study participants. Pregnant women had positive perceptions of using IBCB if the consumption of IBCB was accessible, sustainable, affordable, and provided positive pregnancy outcomes. Therefore, we recommend that nutrition-sensitive agriculture programs should consider the sustainable growing of IBCB in this study rural district, where the majority of the study participants are subsistence farmers. Besides, future studies should investigate the acceptability of IBCB served with other staple foods (cereals, tubers, and plantains), followed by intervention studies to evaluate the effect of IBCB intake on iron status biomarkers and pregnancy outcomes in this study population.”

Round 2

Reviewer 1 Report

Comments and Suggestions for Authors

Since the manuscript has been revised based on the comments, it can be accepted.

Reviewer 2 Report

Comments and Suggestions for Authors

I have carefully reviewed the revised version of the manuscript, as well as the authors’ point-by-point response to the comments and suggestions provided.
The authors have addressed the requested clarifications, particularly regarding the study design, which now enables a clearer understanding of the work.
I believe they have made a significant effort, resulting in a more comprehensive presentation of the study's findings and contributions.

Reviewer 3 Report

Comments and Suggestions for Authors

The authors have substantially improved the manuscript and answered all the questions raised.

Reviewer 4 Report

Comments and Suggestions for Authors

Congratulations for your manuscript